# Generation Enhances Understanding in Unified Multimodal Models via Multi-Representation Generation

**Zihan Su** [1 2 † *]  **Hongyang Wei** [1 *]  **Kangrui Cen** [3 *]  **Yong Wang** [2 ‡]  **Guanhua Chen** [4]
**Chun Yuan** [1]  **Xiangxiang Chu** [2]

## Abstract

Unified Multimodal Models (UMMs) integrate both visual understanding and generation within a single framework. Their ultimate aspiration is to create a cycle where *understanding and generation mutually reinforce each other*. While recent post-training methods have successfully leveraged understanding to enhance generation, the reverse direction of utilizing generation to improve understanding remains largely unexplored. In this work, we propose UniMRG (**Uni**fied **M**ulti-**R**epresentation **G**eneration), a simple yet effective architecture-agnostic post-training method. UniMRG enhances the understanding capabilities of UMMs by incorporating auxiliary generation tasks. Specifically, we train UMMs to generate multiple intrinsic representations of input images, namely *pixel* (reconstruction), *depth* (geometry), and *segmentation* (structure), alongside standard visual understanding objectives. By synthesizing these diverse representations, UMMs capture complementary information regarding appearance, spatial relations, and structural layout. Consequently, UMMs develop a deeper and more comprehensive understanding of visual inputs. Extensive experiments across diverse UMM architectures demonstrate that our method notably enhances fine-grained perception, reduces hallucinations, and improves spatial understanding, while simultaneously boosting generation capabilities.

†Work done during internship at AMAP, Alibaba Group
*Equal contribution ‡Project lead [1]Tsinghua Shenzhen International Graduate School, Tsinghua University [2]AMAP, Alibaba Group [3]Shanghai Jiao Tong University [4]Southern University of Science and Technology. Correspondence to: Yong Wang <wangyong.lz@alibaba-inc.com>, Chun Yuan <yuanc@sz.tsinghua.edu.cn>.

*Proceedings of the 43rd International Conference on Machine Learning*, Seoul, South Korea. PMLR 306, 2026. Copyright 2026 by the author(s).

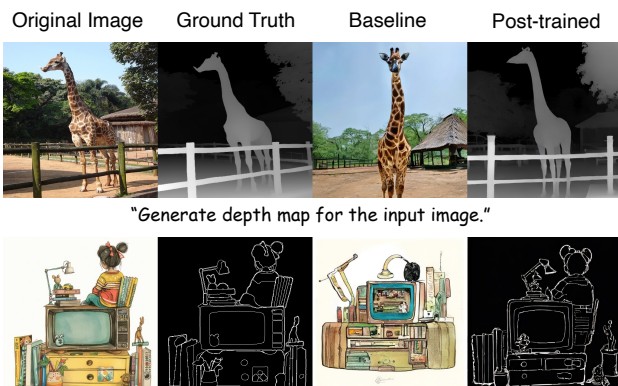

*Figure 1.* **Examples of depth and segmentation generation.** Off-the-shelf UMMs (e.g., Harmon-1.5B (Wu et al., 2025c)) struggle to generate plausible depth/segmentation maps from input images, often producing outputs closer to RGB reconstruction. UniMRG post-trains UMMs to generate these intrinsic representations, encouraging them to internalize geometric cues (depth) and structural cues (segmentation) that are beneficial for visual understanding.

## 1. Introduction

"*What I cannot create, I do not understand.*"

— Richard Feynman

Unified Multimodal Models (UMMs) (Wu et al., 2025c; Xie et al., 2025b; Pan et al., 2025; Deng et al., 2025; Chen et al., 2025a; Wei et al., 2025a) unify visual understanding and generation capabilities in a single architecture, representing a significant advancement in multimodal AI. These models can simultaneously handle visual understanding tasks (Wei et al., 2025b; Yu et al., 2026) and generation tasks (Liu et al., 2025; Chen et al., 2026; Su et al., 2026; Tan et al., 2026), providing greater flexibility and efficiency compared to traditional single-task models. A key aspiration of such unified frameworks is that *understanding and generation capabilities can mutually reinforce each other*, leading to more powerful multimodal intelligence.

Recently, several works have begun to explore the post-training of UMMs. For instance, RecA (Xie et al., 2025a) utilizes rich semantic information from UMM's own understanding encoder for reconstruction, significantly improv-

ing visual generation capability. SRUM (Jin et al., 2025) employs the understanding capability of UMMs to score generated images, enabling self-rewarding training. These works suggest that UMMs' understanding capability can be leveraged to improve generation capability. However, the reverse direction of leveraging generation to improve understanding remains largely unexplored. This leads to a natural question: *can UMMs' generation capabilities also enhance their understanding capabilities?*

To investigate this question, we study a simple yet revealing setup: prompting UMMs to generate *intrinsic visual representations* of an input image beyond RGB appearance. We choose representations that (i) capture *intrinsic factors* that are weakly constrained by pixel reconstruction, (ii) provide *dense* supervision aligned with common failure modes of visual understanding (e.g., spatial reasoning errors and hallucinations). Based on these criteria, we choose *depth* and *segmentation*, leaving other representations for future work. *Depth* explicitly encodes geometry and relative distance, while *segmentation* delineates object boundaries and region partitions, offering an object-centric structural prior. However, as shown in Figure 1, off-the-shelf UMMs often fail to generate such representations, producing outputs that resemble RGB reconstruction rather than plausible depth/segmentation. This suggests that these intrinsic geometric and structural cues are not well captured in UMMs' internal representations. We therefore post-train UMMs to generate depth and segmentation maps, which explicitly encode geometry and structure, respectively. This auxiliary generation encourages UMMs to internalize these regularities, which in turn transfers to stronger understanding.

Figure 2 provides a concrete example of this idea. Focusing on depth as a representative intrinsic signal, off-the-shelf UMMs fail to generate plausible depth maps and also struggle with spatial understanding, whereas image-to-depth post-training enables UMMs to produce coherent depth maps and yields noticeably stronger spatial understanding. Building on this insight, we propose UniMRG (**Uni**fied **M**ulti-**R**epresentation **G**eneration), a simple yet effective architecture-agnostic post-training method that improves understanding via auxiliary generation of intrinsic representations. Specifically, UniMRG trains UMMs to generate multiple intrinsic image representations, including *pixel* (reconstruction), *depth* (geometry), and *segmentation* (structure), along with standard visual understanding objectives. By learning to synthesize these complementary representations, UMMs can better capture appearance cues, spatial relations, and structural layout, leading to stronger and more comprehensive visual understanding.

We validate our approach across various UMM architectures, including autoregressive, masked autoregressive, and diffusion-based UMMs. Experimental results consistently

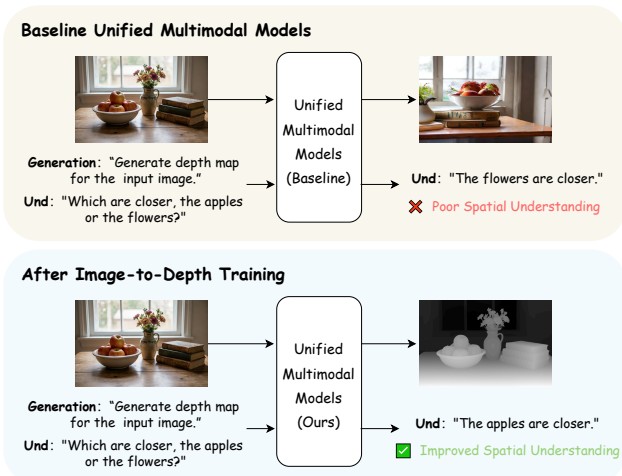

*Figure 2.* **Motivation: Intrinsic visual representation generation enhances visual understanding.** Top: Off-the-shelf UMMs fail to generate plausible depth maps and struggle with spatial understanding. Bottom: After image-to-depth training, UMMs generate coherent depth maps and exhibit stronger spatial understanding, correctly identifying spatial relationships.

show that our method notably improves fine-grained perception, mitigates hallucinations, and strengthens spatial understanding, while also enhancing generation performance. Our contributions are summarized as follows:

- We propose UniMRG, a simple yet effective architecture-agnostic post-training method for UMMs that leverages generation capabilities to enhance understanding.

- We introduce a multi-representation generation strategy that trains UMMs to generate multiple intrinsic image representations, namely *pixel* (reconstruction), *depth* (geometry), and *segmentation* (structure), alongside standard visual understanding objectives, thereby capturing diverse visual information to enhance understanding.

- Extensive experiments across different UMM architectures show that our method notably enhances fine-grained perception, reduces hallucinations, and improves spatial understanding, while simultaneously boosting generation capabilities.

## 2. Related Work

**Unified Multimodal Models (UMMs).** Unified multimodal models can be categorized into three architectural paradigms: **(1) AR.** Models like Janus-Pro (Chen et al., 2025b) and Show-o (Xie et al., 2025b) encode images into discrete tokens via VQ-VAE (Van Den Oord et al., 2017), enabling autoregressive prediction similar to text tokens. **(2) AR+MAR.** Models like Harmon (Wu et al., 2025c) leverage Masked Autoregressive (MAR) (Li et al., 2024b) modeling, using MAR features as unified representations. **(3) AR+Diffusion.** Models such as OpenUni (Wu

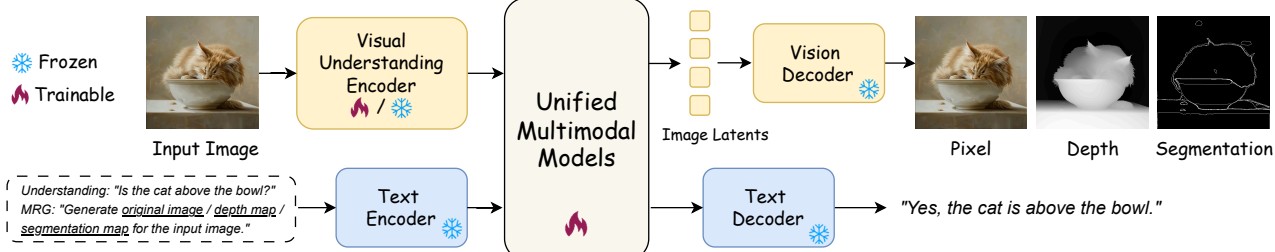

*Figure 3.* **Overview of UniMRG.** The input image is fed into the visual understanding encoder, and the UMM is jointly trained on four tasks: (1) *Image reconstruction*: reconstructing the input image to enhance generation capabilities. (2) *Image-to-depth*: generating depth maps to learn geometric cues and spatial relations. (3) *Image-to-segmentation*: generating segmentation maps to learn structural cues and region partitions. (4) *Image understanding*: performing standard vision-language understanding tasks. The understanding encoder is updated for UMMs with a shared encoder for generation and understanding; otherwise it is frozen.

et al., 2025b) and BLIP-3-o (Chen et al., 2025a) condition diffusion decoders on MLLM hidden states, while models like BAGEL (Deng et al., 2025) adopt the Mixture-of-Transformer-Experts (MoT) architecture to jointly perform understanding and generation.

**Post-training for UMMs.** Recently, several works have begun exploring post-training for UMMs. RecA (Xie et al., 2025a) reconstructs images by exploiting semantic representations extracted from the UMM's understanding encoder, which markedly enhances its generation ability. SRUM (Jin et al., 2025) turns the UMM's understanding capacity into a scoring signal for synthesized images, supporting a self-rewarding training paradigm. Han et al. (2025) propose an internal gap-based self-improvement framework, which mitigates internal gaps in UMMs by leveraging understanding to guide generation. UniCorn (Han et al., 2026) distills latent understanding into explicit generative signals in UMMs. *However, existing works primarily focus on leveraging UMMs' understanding capabilities to enhance generation, while we explore using UMMs' generation capabilities to improve understanding.*

**Depth Estimation and Segmentation.** *Depth estimation* is a fundamental dense prediction task that infers per-pixel scene geometry and relative distance. Early work studies supervised or self-supervised monocular depth prediction (Godard et al., 2019), while later works emphasize stronger cross-dataset generalization via multi-dataset training (Ranftl et al., 2020). Recent models such as Depth Anything V2 (Yang et al., 2024) further improve robustness and generalization, making depth a practical representation for capturing spatial relations. *Segmentation* provides region-level scene decomposition (Long et al., 2015; Chen et al., 2018; Zhang et al., 2022) via semantic segmentation, as well as instance-level masks (He et al., 2017) for object-wise partitioning. Beyond category-aware semantic/instance segmentation, Segment Anything (SAM) (Kirillov et al., 2023) enables general-purpose, instance-like mask generation, offering strong structural cues about object boundaries and

region partitions. *Motivated by the complementarity of these two representations, UniMRG treats depth maps and segmentation maps as auxiliary generation targets, enabling UMMs to acquire richer geometric and structural knowledge that transfers to downstream understanding.*

## 3. Unified Multi-Representation Generation

We propose UniMRG (**Uni**fied **M**ulti-**R**epresentation **G**eneration), a simple yet effective post-training method that enhances UMMs' understanding capabilities via auxiliary generation tasks. In this section, we first detail our multi-representation generation strategy in Section 3.1. Then, we provide further discussion and insights in Section 3.2. Preliminaries on different generation paradigms of UMMs are provided in Appendix A. The overall pipeline is illustrated in Figure 3.

### 3.1. Multi-Representation Generation Strategy

Most existing post-training recipes for UMMs primarily exploit *understanding* signals to improve *generation* (Han et al., 2025; Jin et al., 2025). In contrast, we study the reverse direction: *can generation improve visual understanding?* We answer this by leveraging the input image's *intrinsic visual representations* as auxiliary generation targets. Consequently, UniMRG trains UMMs on four tasks simultaneously: image reconstruction, image-to-depth, image-to-segmentation, and image understanding.

**Multi-Task Objective.** We use $f$ to represent the UMM, $h$ for the visual embedding of the input image extracted by the UMM's understanding encoder, and $\mathcal{L}(\cdot, \cdot)$ for the loss function. For understanding tasks, $\mathcal{L}(\cdot, \cdot)$ is the cross-entropy loss. For generation tasks, $\mathcal{L}(\cdot, \cdot)$ is the diffusion loss for diffusion-based generation models or the cross-entropy loss for autoregressive generation models. The individual loss terms are defined as follows.

*(1) The visual understanding loss* improves the model's

understanding capabilities through standard vision-language tasks:

$$\mathcal{L}_{\text{und}} = \mathcal{L}(f(h, t_{\text{question}}), t_{\text{answer}}) \quad (1)$$

where $t_{\text{question}}$ is the question prompt and $t_{\text{answer}}$ is the answer.

*(2) The image reconstruction loss* enhances the UMMs' generation capabilities:

$$\mathcal{L}_{\text{pixel}} = \mathcal{L}(f(h, t_{\text{pixel}}), I_{\text{pixel}}) \quad (2)$$

where $t_{\text{pixel}}$ is the prompt *"Generate original image for the input image"* and $I_{\text{pixel}}$ is the input image (for simplicity, we omit the VAE decoding process here and in subsequent formulations).

*(3) The image-to-depth loss* compels UMMs to learn geometric cues and spatial relations:

$$\mathcal{L}_{\text{depth}} = \mathcal{L}(f(h, t_{\text{depth}}), I_{\text{depth}}) \quad (3)$$

where $t_{\text{depth}}$ is the prompt *"Generate depth map for the input image"* and $I_{\text{depth}}$ is the target depth map preprocessed from the input image using Depth Anything V2 (Yang et al., 2024).

*(4) The image-to-segmentation loss* compels UMMs to learn structural cues and region partitions:

$$\mathcal{L}_{\text{seg}} = \mathcal{L}(f(h, t_{\text{seg}}), I_{\text{seg}}) \quad (4)$$

where $t_{\text{seg}}$ is the prompt *"Generate segmentation map for the input image"* and $I_{\text{seg}}$ is the target segmentation map preprocessed from the input image using automatic mask generation with Segment Anything (Kirillov et al., 2023).

The overall training objective of UniMRG combines these four loss terms:

$$\mathcal{L}_{\text{total}} = \lambda_{\text{pixel}}\mathcal{L}_{\text{pixel}} + \lambda_{\text{depth}}\mathcal{L}_{\text{depth}} + \lambda_{\text{seg}}\mathcal{L}_{\text{seg}} \\ + \lambda_{\text{und}}\mathcal{L}_{\text{und}} \quad (5)$$

where $\lambda_{\text{pixel}}$, $\lambda_{\text{depth}}$, $\lambda_{\text{seg}}$, and $\lambda_{\text{und}}$ are the weights for each loss term. In our experiments, we set all weights to 1.

**Training and Inference Details.** We freeze the VQ-VAE (Van Den Oord et al., 2017) and text encoder/decoder during training. The understanding encoder is updated for UMMs with a shared encoder for generation and understanding (e.g., Harmon (Wu et al., 2025c)); otherwise it is frozen. All other components of the UMM are trainable. During inference, UniMRG operates identically to standard UMMs without any architectural modifications or additional computational overhead.

### 3.2. Discussion

**Why Intrinsic Visual Representations Are Necessary.** To improve understanding via generation, a natural starting

point is image reconstruction, which strengthens appearance modeling and has been shown effective for boosting generation (Xie et al., 2025a). However, pixel-level supervision is dominated by textures and colors and provides only weak constraints on intrinsic factors such as geometry and structure. As a result, reconstruction alone is often insufficient to induce the *intrinsic* cues needed for challenging understanding scenarios, such as fine-grained perception, spatial understanding, and hallucination reduction. We therefore supervise UMMs to generate *intrinsic visual representations* as auxiliary targets, explicitly encouraging the model to capture complementary factors beyond RGB appearance.

**Which Intrinsic Representations Are Suitable?** Representations that capture intrinsic factors of a scene beyond RGB appearance are suitable. In this work, we focus on depth and segmentation representations. The exploration of more representations is left for future work. *Depth* maps expose scene geometry and relative distance, directly supporting spatial relations (e.g., front/behind, near/far). This directly benefits spatial understanding and reasoning in downstream VQA. *Segmentation* maps delineate object boundaries and region partitions, providing an object-centric structural prior that helps disentangle entities and reduces spurious attribute binding, a common source of hallucinations.

**Differences from Prior Work.** *(1) Reconstruction for UMMs.* RecA (Xie et al., 2025a) uses UMM understanding representations to guide reconstruction, thereby improving generation, whereas UniMRG studies the reverse direction by using auxiliary generation to strengthen visual understanding. *(2) Reconstruction for Understanding.* DIVA (Wang et al., 2024b) and ROSS (Wang et al., 2024a) show that reconstruction-style objectives in pixel space or VAE latent space can benefit understanding; in contrast, UniMRG emphasizes *intrinsic* visual representations beyond appearance-level reconstruction. *(3) Representation Alignment.* REPA (Yu et al., 2024) aligns generative features to pretrained encoders to improve generative training, while UniMRG instead uses intrinsic-representation generation for better understanding. *(4) Depth Cues for Understanding.* SpatialRGPT (Cheng et al., 2024) and DepthVLA (Yuan et al., 2025) enhance spatial understanding in *VLM* by incorporating depth cues via *additional modules*. In contrast, UniMRG targets the *UMM* setting and introduces *no architectural changes* to the UMM.

## 4. Experiments

### 4.1. Experimental Setup

**Model Architectures.** We validate UniMRG on three representative UMM architectures with different generation paradigms:

- **AR:** Show-o-1.3B (Xie et al., 2025b) is a unified trans-

*Table 1.* **Comparison of UniMRG with other post-training methods for UMMs.** The green row shows the Gain (↑) over the base model. SFT denotes supervised fine-tuning using only the visual understanding loss.

| Method | General | Fine-Grained | Hallucination | Spatial Understanding | | Generation | |
| --- | --- | --- | --- | --- | --- | --- | --- |
| | MMBench | MMVP | Hallusion | RWQA | VSR | GenEval | DPG |
| *Harmon-1.5B* | | | | | | | |
| Base | 50.43 | 60.00 | 46.69 | 46.67 | 60.88 | 71.37 | 80.52 |
| SFT | 50.43 | 60.00 | 47.95 | 50.20 | 60.88 | 0.30 | 2.85 |
| RecA | 49.50 | 61.00 | 47.11 | 49.54 | 59.00 | 83.86 | **86.61** |
| UniMRG | **52.23** | **62.67** | **49.32** | **51.90** | **61.21** | **85.26** | 85.27 |
| Gain (↑) | +1.80 | +2.67 | +2.63 | +5.23 | +0.33 | +13.89 | +4.75 |
| *OpenUni-3.6B* | | | | | | | |
| Base | 81.19 | 71.67 | 60.88 | 65.23 | 66.69 | 61.13 | 79.41 |
| SFT | 79.81 | 73.33 | 64.25 | 63.14 | 72.18 | 46.03 | 70.24 |
| RecA | 81.19 | 71.67 | 60.88 | 65.23 | 66.69 | **70.07** | 81.54 |
| UniMRG | **81.44** | **74.67** | **64.56** | **66.01** | **73.90** | 68.00 | **81.78** |
| Gain (↑) | +0.25 | +3.00 | +3.68 | +0.78 | +7.21 | +6.87 | +2.37 |
| *Show-o-1.3B* | | | | | | | |
| Base | 47.85 | 50.00 | 46.06 | 38.17 | 54.26 | 67.32 | 81.94 |
| SFT | 48.02 | 50.67 | 45.85 | 37.65 | 51.55 | 67.09 | 81.98 |
| RecA | 47.42 | 47.33 | 46.69 | 36.21 | 52.70 | **71.52** | 84.44 |
| UniMRG | **48.11** | **51.33** | **47.00** | **39.87** | **54.42** | 71.40 | **84.55** |
| Gain (↑) | +0.26 | +1.33 | +0.94 | +1.70 | +0.16 | +4.08 | +2.61 |

former that generates 512×512 images through autoregressive masked prediction. It processes text tokens autoregressively while handling image tokens via masked prediction. We use the CLIP variant in our experiments.

- **AR+MAR:** Harmon-1.5B (Wu et al., 2025c) employs a shared Masked Autoregressive (MAR) (Li et al., 2024b) encoder for both visual generation and comprehension, generating 384×384 images. MAR learns rich semantics through mask-and-reconstruct processes, enabling consistent semantic grounding across tasks.

- **AR+Diffusion:** OpenUni-3.6B (Wu et al., 2025b) serves as an open-source implementation following the MetaQueries (Pan et al., 2025) architecture. It bridges a frozen multimodal LLM (i.e., InternVL3-2B (Zhu et al., 2025)) with a diffusion generator (i.e., SANA-1.6B (Xie et al., 2024)) using learnable queries and a lightweight transformer-based connector, generating 512×512 images.

**Implementation Details.** We use the image understanding datasets LLaVA Mix-665K (Liu et al., 2023b) and LLaVA-Next-Data (Liu et al., 2024a). Our experiments are conducted on 8 NVIDIA H20 GPUs. Notably, our method is *resource-efficient*, requiring only about 3 hours for OpenUni, 5 hours for Harmon, and 8 hours for Show-o. For more implementation details, please refer to Appendix B.

**Evaluation Metrics.** For generation capabilities, we eval-

uate UniMRG on GenEval (Ghosh et al., 2023) and DPG-Bench (Hu et al., 2024). For understanding capabilities, we evaluate UniMRG on (i) *general understanding*: MM-Bench (Liu et al., 2024b) English dev split, (ii) *fine-grained perception*: MMVP (Tong et al., 2024), (iii) *hallucination*: HallusionBench (Guan et al., 2024), and (iv) *spatial understanding*: RealWorldQA (RWQA) (xAI, 2024) and Visual Spatial Reasoning (VSR) (Liu et al., 2023a). All understanding evaluations are conducted with VLMEvalKit (Duan et al., 2024).

### 4.2. Main Results

**Comparison with UMM Post-training Methods.** We compare UniMRG with SFT (i.e., training with only image understanding loss) and RecA (Xie et al., 2025a) (which performs image reconstruction using semantic features) across different UMM architectures. Results are shown in Table 1, and detailed generation benchmark metrics are provided in Table 4 in the Appendix. Notably, SFT, which only trains UMMs on understanding tasks, leads to a dramatic decline in generation metrics. For instance, on Harmon, GenEval drops from 71.37 to 0.30, and DPGBench from 80.52 to 2.85 after SFT. While RecA markedly improves generation capabilities, it provides no gains for understanding.

In contrast, UniMRG achieves remarkable improvements in both understanding and generation capabilities. For un-

*Table 2.* **Comparison with state-of-the-arts on visual understanding benchmarks.** Our model is OpenUni post-trained with UniMRG.

| Type | Model | # LLM Params | General | Fine-Grained | Hallucination | Spatial Understanding | |
|---|---|---|---|---|---|---|---|
| | | | MMBench | MMVP | Hallusion | RWQA | VSR |
| *Und. Only* | LLaVA-OV (Li et al., 2024a) | 0.5B | 55.15 | 53.67 | 51.95 | 46.41 | 51.47 |
| | InternVL3 (Zhu et al., 2025) | 2B | 81.19 | 71.67 | 60.88 | 65.23 | 66.69 |
| | Qwen2.5-VL (Bai et al., 2025b) | 3B | 78.26 | 64.67 | – | – | – |
| | Qwen3-VL (Bai et al., 2025a) | 2B | 77.32 | 72.00 | – | – | – |
| | Emu3-Chat (Wang et al., 2024c) | 8B | 64.35 | 67.00 | 52.16 | 56.99 | 71.93 |
| *Unified* | Chameleon (Team, 2024) | 7B | 35.70 | 0.00 | – | – | – |
| | Janus (Wu et al., 2025a) | 1.3B | 53.52 | 56.67 | 50.16 | 43.66 | 61.37 |
| | Janus-Pro (Chen et al., 2025b) | 7B | 66.67 | 63.00 | 60.15 | 41.83 | 71.03 |
| | Harmon (Wu et al., 2025c) | 1.5B | 50.43 | 60.00 | 46.69 | 46.67 | 60.88 |
| | Show-o (Xie et al., 2025b) | 1.3B | 47.85 | 50.00 | 46.06 | 38.17 | 54.26 |
| | BAGEL (Deng et al., 2025) | 3B | 79.20 | 54.70 | – | – | – |
| | UniLIP (Tang et al., 2025) | 2B | 80.70 | 73.00 | 60.57 | 64.18 | 65.55 |
| | OpenUni (Wu et al., 2025b) | 2B | 81.19 | 71.67 | 60.88 | 65.23 | 66.69 |
| | UniMRG (Ours) | 2B | **81.44** | **74.67** | **64.56** | **66.01** | **73.90** |

derstanding metrics, UniMRG achieves state-of-the-art results across different UMM architectures, notably enhancing UMMs' fine-grained perception, hallucination mitigation, and spatial understanding capabilities. For example, on OpenUni-3.6B, MMVP improves from 71.67 to 74.67 (+3.00), HallusionBench from 60.88 to 64.56 (+3.68), and VSR from 66.69 to 73.90 (+7.21), demonstrating notable improvements across all three capabilities. For generation metrics, UniMRG achieves results comparable to RecA, significantly improving UMMs' generation capabilities. Additionally, Show-o shows smaller improvements on understanding metrics, which we analyze in Section 4.4.

**Comparison with State-of-the-Arts.** We compare OpenUni post-trained with UniMRG against state-of-the-art UMMs and understanding-only models of similar scale. Results are shown in Table 2. Our method achieves state-of-the-art results in fine-grained perception, hallucination mitigation, and spatial understanding, evidencing the effectiveness of UniMRG. For example, on HallusionBench, UniMRG achieves 64.56, significantly outperforming the second-best result of 60.88.

**Qualitative Results.** Qualitative generation results are shown in Figure 4a, and understanding results are shown in Figure 4b. For generation capabilities, UMMs post-trained with UniMRG better follow prompts involving multiple objects, spatial relationships, and complex attributes. For example, given the prompt *"a photo of a pink skateboard and a black train"*, the baseline generates a train with mixed red and black colors, while our method correctly generates a black train. For understanding capabilities, UMMs post-trained with UniMRG exhibit improved fine-grained perception, fewer hallucinations, and stronger spatial understanding. For example, on the spatial question *"Which is closer?"*, the baseline answers incorrectly, whereas UniMRG gives the correct answer. This indicates that the

image-to-depth objective encourages the model to internalize depth-related cues and relative spatial ordering, making distance comparisons more reliable. On the hallucination check *"Is there any yogurt in this figure?"*, the baseline answers *"yes"* by fabricating a non-existent object, while UniMRG correctly answers *"no"*. This improvement is consistent with the image-to-segmentation objective, which emphasizes object boundaries and region-level scene decomposition, helping the model better ground object presence in the visual content.

### 4.3. Ablation Study

**Quantitative Ablation Results.** The quantitative ablation study results on Harmon-1.5B are shown in Table 3. After SFT (only understanding loss), generation capabilities dramatically decline, with GenEval dropping from 71.37 to 0.30, indicating that training solely on understanding tasks severely degrades UMM's generation capabilities. After adding pixel representation generation, generation capabilities considerably improve. However, understanding capabilities show no gains at this point. After adding depth representation generation, understanding capabilities notably improve. For example, for spatial understanding, VSR improves from 59.00 to 60.39. After adding segmentation representation generation, understanding capabilities continue to improve. For example, for hallucination detection, Hallusion improves from 48.26 to 49.32. The results show that adding depth and segmentation representation generation markedly improves understanding capabilities, while not harming generation capabilities (GenEval: 83.86→85.26, DPGBench: 86.61→85.27).

**Qualitative Ablation Results.** We explore the impact of different representation generations on the quality of UMM-generated images, with results shown in Figure 5. With

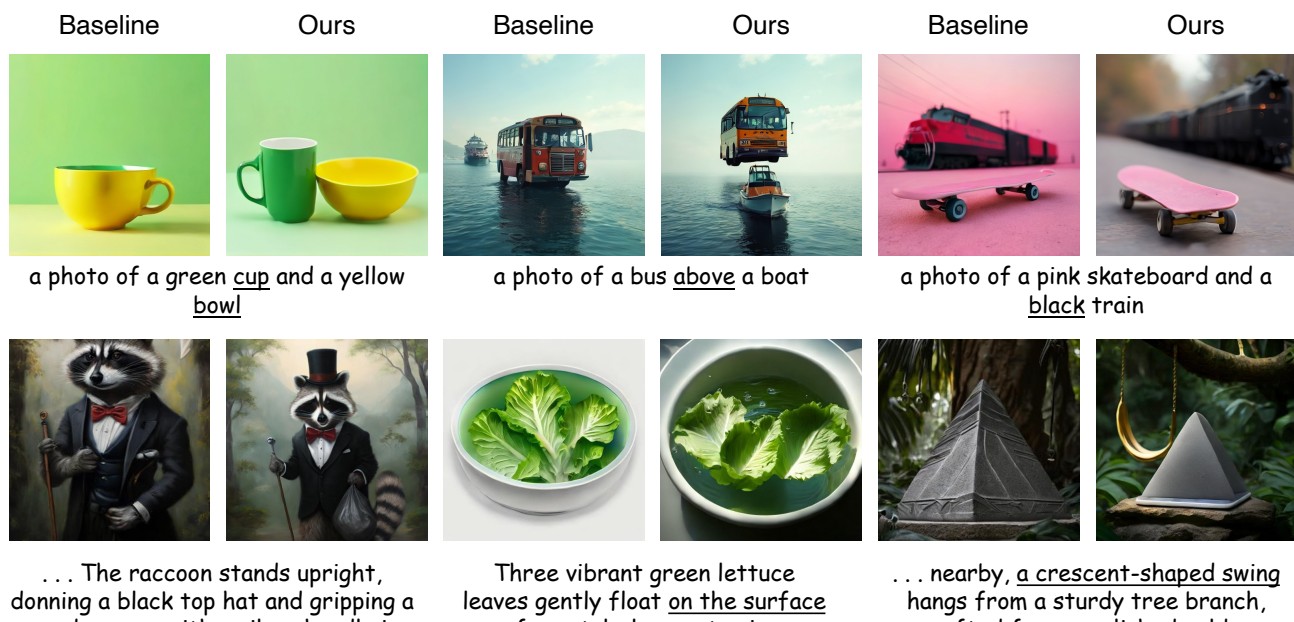

*(a)* **Qualitative generation results.** UMMs post-trained with UniMRG better follow prompts involving multiple objects, spatial relationships, and complex attributes.

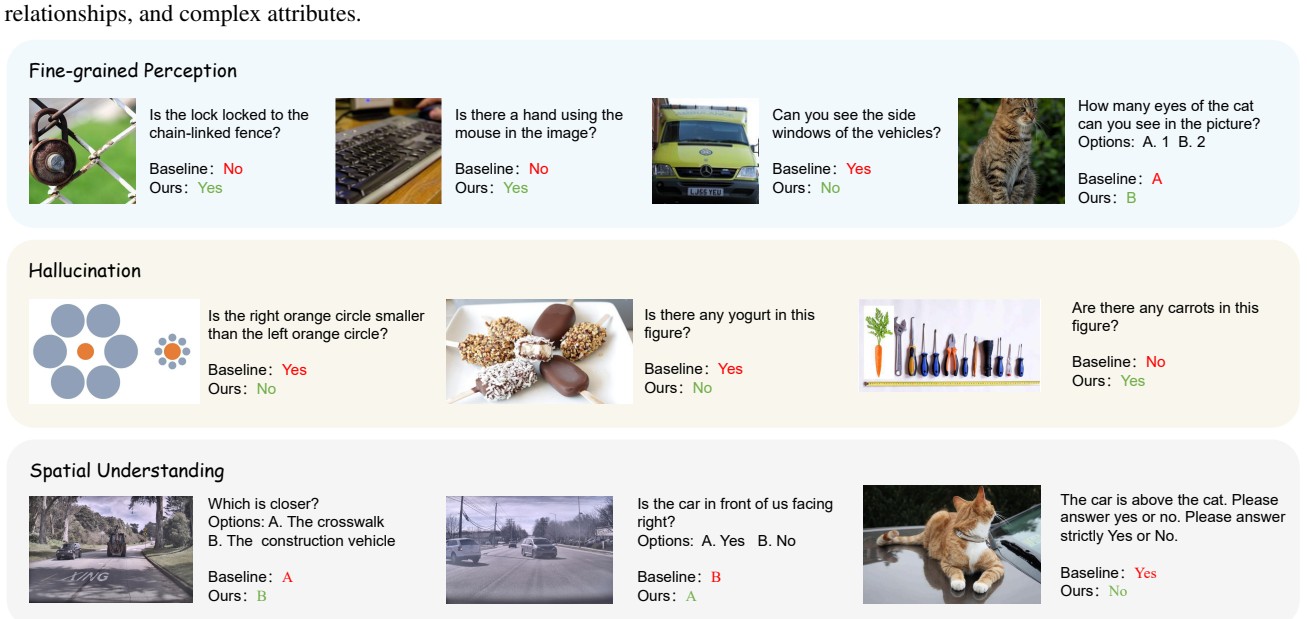

*(b)* **Qualitative understanding results.** UMMs post-trained with UniMRG exhibit improved fine-grained perception, reduced hallucinations, and enhanced spatial understanding.

*Figure 4.* **Qualitative results on generation and understanding.**

only SFT for image understanding, the quality of UMM-generated images severely degrades, resulting in meaningless noise. With only depth representation generation, the generated image distribution becomes biased toward depth maps, leading to darkened images with loss of texture details. Similarly, with only segmentation representation generation, the generated image distribution also becomes biased toward

segmentation maps rather than natural images. However, after adding pixel representation generation on top of depth and segmentation representation generation, the quality of UMM-generated images notably improves, producing images that closely resemble natural images. This indicates that pixel representation generation is essential for maintaining the quality of UMM-generated images.

*Table 3.* **Quantitative Ablation Results.**

| Method | General | Fine-Grained | Hallucination | Spatial Understanding | | Generation | |
|---|---|---|---|---|---|---|---|
| | MMBench | MMVP | Hallusion | RWQA | VSR | GenEval | DPG |
| Harmon-1.5B | 50.43 | 60.00 | 46.69 | 46.67 | 60.88 | 71.37 | 80.52 |
| + Visual Understanding | 50.43 | 60.00 | 47.95 | 50.20 | 60.88 | 0.30 | 2.85 |
| + Pixel Generation | 49.50 | 61.00 | 47.11 | 49.54 | 59.00 | 83.86 | **86.61** |
| + Depth Generation | 50.60 | 62.00 | 48.26 | 50.46 | 60.39 | 85.15 | 84.96 |
| + Segmentation Generation | **52.23** | **62.67** | **49.32** | **51.90** | **61.21** | **85.26** | 85.27 |

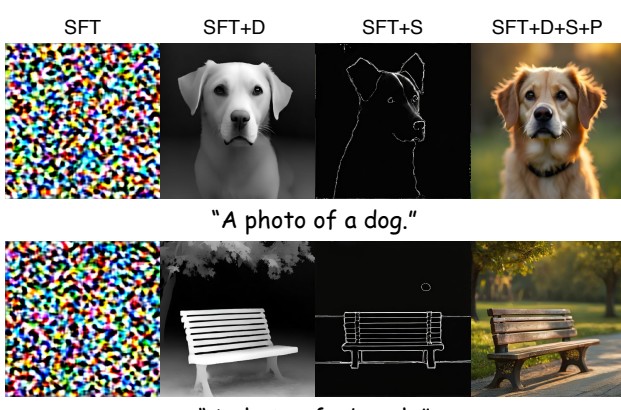

"A photo of a dog."

"A photo of a bench."

*Figure 5.* **Qualitative ablation study on the quality of UMM-generated images with different representation generations.** D denotes depth representation generation, S denotes segmentation representation generation, and P denotes pixel representation generation.

## 4.4. Discussion

We investigate two questions: *(i) whether UniMRG induces genuine intrinsic representation generation ability that internalizes regularities useful for visual understanding, rather than merely overfitting to the training data distribution,* and *(ii) whether stronger intrinsic representation generation correlates with gains in understanding.*

To answer these questions, we evaluate depth representation generation on MidjourneyV6 (CortexLM, 2024), a *synthetic* image dataset whose distribution differs substantially from the *real* image distribution used for training. We randomly sample 1,000 images and compare three UMMs using their original weights and the corresponding UniMRG-posttrained weights. For each model, we prompt it to generate a depth representation and measure its similarity to targets predicted by Depth Anything V2. We measure similarity using $1 - \mathrm{MAE}$, where $\mathrm{MAE}$ denotes the *mean absolute error* between the predicted and target depth maps (higher is better). Results are shown in Figure 6.

UniMRG substantially improves depth generation ability for Harmon and OpenUni, increasing from 0.623 to 0.822 and from 0.617 to 0.834, respectively. This shows that the enhanced depth representation generation generalizes well

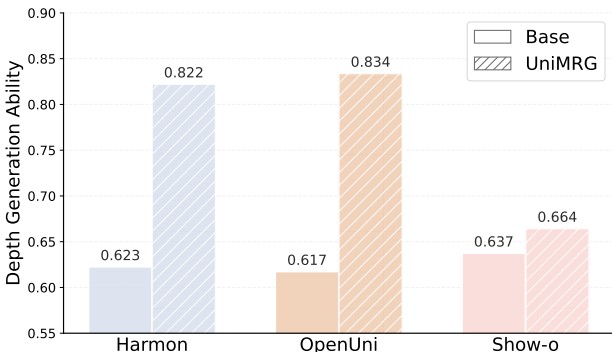

*Figure 6.* **OOD depth representation generation on MidjourneyV6.** We sample 1,000 images from MidjourneyV6 and measure the similarity between model-generated depth maps and Depth Anything V2 targets using $1 - \mathrm{MAE}$ (higher is better), comparing base and UniMRG weights.

to synthetic, out-of-distribution images. In contrast, Show-o exhibits only a marginal gain ($0.637 \rightarrow 0.664$). We attribute this to a representational bottleneck: Show-o relies on a VQ codebook with only 4,096 tokens, which limits its expressive capacity for jointly generating multiple representations (pixel, depth, and segmentation). Consequently, UniMRG yields only marginal improvements in understanding capability on Show-o. Similar modest gains due to capacity constraints in Show-o are also observed in RecA (Xie et al., 2025a). These results suggest that when the generative representation space is limited (e.g., by a small VQ codebook), producing reliable intrinsic targets becomes difficult, which may bound the gains in understanding performance.

## 5. Conclusion

In this work, we explore improving understanding via generation in Unified Multimodal Models (UMMs). We propose UniMRG, a simple yet effective architecture-agnostic posttraining method that trains UMMs with auxiliary generation of intrinsic visual representations, including *pixel*, *depth*, and *segmentation*, alongside standard understanding objectives. Synthesizing these representations encourages UMMs to internalize geometric and structural regularities that improve understanding. Across diverse UMM architectures, UniMRG consistently improves fine-grained perception, re-

duces hallucinations, and strengthens spatial understanding, while also enhancing generation quality. Looking forward, we will extend UniMRG to more intrinsic representations (e.g., pose, sketches) and to video settings. We hope this work motivates further research on synergies between understanding and generation in multimodal models.

## Acknowledgments

This work is supported by the SSTIC Grant (KJZD20230923115106012, KJZD20230923114916032, GJHZ20240218113604008).

## Impact Statement

This paper studies how auxiliary generation of intrinsic visual representations (pixel, depth, and segmentation) can improve visual understanding in unified multimodal models, leading to better fine-grained perception, reduced hallucinations, and stronger spatial understanding. These improvements may have positive societal impact by making multimodal systems more reliable for applications that require accurate grounding (e.g., assistive tools, education, human-computer interaction, and downstream decision-support settings where hallucinations are harmful).

At the same time, strengthening generation capabilities can also increase the risk of misuse of image generation (e.g., creating deceptive or misleading synthetic content). Our method does not by itself provide a complete solution to these risks; responsible deployment should include appropriate safeguards such as content provenance and policy-based filtering for generated media, careful evaluation across diverse data to mitigate bias, and human oversight in high-stakes use cases.

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

# A. Preliminaries

In this section, we provide a brief introduction for the three primary generative paradigms utilized in Unified Multimodal Models (UMMs): Autoregressive (AR) modeling, Diffusion Modeling, and Masked Autoregressive (MAR) modeling.

## A.1. Autoregressive Modeling

Visual Autoregressive models typically operate on a discrete latent space derived from a VQ-VAE (Van Den Oord et al., 2017). Let $\boldsymbol{x} = [x_1, \ldots, x_T]$ denote the flattened sequence of discrete tokens from a codebook. Standard AR models decompose the joint distribution via the chain rule of probability. The model maximizes the log-likelihood:

$$\log p_\theta(\boldsymbol{x}) = \sum_{t=1}^{T} \log p_\theta(x_t \mid x_{<t}), \tag{6}$$

where $x_{<t}$ denotes the causal context. The training objective is to minimize the negative log-likelihood (NLL):

$$\mathcal{L}_{\text{Causal}}(\theta) = -\mathbb{E}_{\boldsymbol{x} \sim p_{data}} \left[ \sum_{t=1}^{T} \log p_\theta(x_t \mid x_{<t}) \right]. \tag{7}$$

To enforce the autoregressive property, a causal mask $\boldsymbol{M}$ is applied to the self-attention mechanism, such that the attention logits satisfy $A_{ij} = -\infty$ if $j > i$, ensuring strict sequential dependency.

To enable parallel token prediction and utilize bidirectional context, MaskGit (Chang et al., 2022) employs a masked prediction strategy. A binary mask $\boldsymbol{m} \in \{0,1\}^T$ is sampled, where $m_i = 1$ indicates a visible token and $m_i = 0$ a masked token. The model predicts the masked tokens $\boldsymbol{x}_{\bar{\boldsymbol{m}}}$ conditioned on *all* visible tokens $\boldsymbol{x}_{\boldsymbol{m}}$ (and the mask pattern) simultaneously. The objective minimizes the cross-entropy loss on the masked positions:

$$\mathcal{L}_{\text{MaskGit}}(\theta) = -\mathbb{E}_{\boldsymbol{x},\boldsymbol{m}} \left[ \sum_{i \in \bar{\boldsymbol{m}}} \log p_\theta(x_i \mid \boldsymbol{x}_{\boldsymbol{m}}) \right]. \tag{8}$$

Unlike standard AR, this approach uses unmasked bidirectional attention, allowing the model to attend to both past and future tokens. During inference, it employs iterative decoding to generate tokens in parallel steps based on confidence scores.

UMMs like Chameleon (Team, 2024), Janus-Pro (Chen et al., 2025b), and Show-o (Xie et al., 2025b) adopt autoregressive generation paradigms.

## A.2. Diffusion Modeling

This paradigm generates data by reversing a continuous corruption process. We detail the stochastic framework (DDPM) and the deterministic ODE framework (Flow Matching).

**Denoising Diffusion Probabilistic Models (DDPM).** DDPM (Ho et al., 2020) defines a forward diffusion process that gradually adds Gaussian noise to the data $\boldsymbol{x}_0 \sim p_{data}$ over $T$ timesteps. The transition kernel $q(\boldsymbol{x}_t \mid \boldsymbol{x}_{t-1})$ is a Gaussian:

$$q(\boldsymbol{x}_t \mid \boldsymbol{x}_{t-1}) = \mathcal{N}(\boldsymbol{x}_t; \sqrt{1-\beta_t}\boldsymbol{x}_{t-1}, \beta_t \mathbf{I}), \tag{9}$$

where $\beta_t$ is a variance schedule. A key property allows sampling $\boldsymbol{x}_t$ directly from $\boldsymbol{x}_0$. Let $\alpha_t = 1 - \beta_t$ and $\bar{\alpha}_t = \prod_{s=1}^{t} \alpha_s$, the marginal distribution is:

$$q(\boldsymbol{x}_t \mid \boldsymbol{x}_0) = \mathcal{N}(\boldsymbol{x}_t; \sqrt{\bar{\alpha}_t}\boldsymbol{x}_0, (1-\bar{\alpha}_t)\mathbf{I}). \tag{10}$$

Using the reparameterization trick, we can express $\boldsymbol{x}_t = \sqrt{\bar{\alpha}_t}\boldsymbol{x}_0 + \sqrt{1-\bar{\alpha}_t}\boldsymbol{\epsilon}$, where $\boldsymbol{\epsilon} \sim \mathcal{N}(\mathbf{0}, \mathbf{I})$.

The reverse process $p_\theta(\boldsymbol{x}_{t-1} \mid \boldsymbol{x}_t)$ aims to recover the data. It is modeled as a Gaussian with learnable mean $\boldsymbol{\mu}_\theta$ and fixed variance $\tilde{\beta}_t$:

$$p_\theta(\boldsymbol{x}_{t-1} \mid \boldsymbol{x}_t) = \mathcal{N}(\boldsymbol{x}_{t-1}; \boldsymbol{\mu}_\theta(\boldsymbol{x}_t, t), \tilde{\beta}_t \mathbf{I}). \tag{11}$$

Ideally, $\boldsymbol{\mu}_\theta$ should predict the posterior mean. We parameterize $\boldsymbol{\mu}_\theta$ via a noise prediction network $\boldsymbol{\epsilon}_\theta(\boldsymbol{x}_t, t)$:

$$\boldsymbol{\mu}_\theta(\boldsymbol{x}_t, t) = \frac{1}{\sqrt{\alpha_t}}\left(\boldsymbol{x}_t - \frac{\beta_t}{\sqrt{1 - \bar{\alpha}_t}}\boldsymbol{\epsilon}_\theta(\boldsymbol{x}_t, t)\right). \tag{12}$$

The simplified training objective is to minimize the error of the noise prediction:

$$\mathcal{L}_{\text{DDPM}}(\theta) = \mathbb{E}_{\boldsymbol{x}_0, t, \boldsymbol{\epsilon}}\left[\|\boldsymbol{\epsilon} - \boldsymbol{\epsilon}_\theta(\boldsymbol{x}_t, t)\|^2\right]. \tag{13}$$

**Flow Matching (FM).** Flow Matching (Lipman et al., 2022) models the generative process via a Continuous Normalizing Flow (CNF). We define a probability path $p_t(\boldsymbol{x})$ that interpolates between a source noise distribution $p_0(\boldsymbol{x}) = \mathcal{N}(\boldsymbol{0}, \boldsymbol{I})$ and the target data distribution $p_1(\boldsymbol{x}) \approx p_{\text{data}}$. This flow is governed by an Ordinary Differential Equation (ODE):

$$\frac{d\boldsymbol{x}}{dt} = \boldsymbol{v}_t(\boldsymbol{x}), \quad \boldsymbol{x}(0) \sim p_0, \tag{14}$$

where $\boldsymbol{v}_t(\cdot)$ is a time-dependent vector field. The goal is to learn a neural network $\boldsymbol{v}_\theta(\boldsymbol{x}, t)$ that approximates $\boldsymbol{v}_t$.

To make training tractable, we use *Conditional Flow Matching* (CFM). During training, we sample a source-target pair $(\boldsymbol{x}_0, \boldsymbol{x}_1)$ with $\boldsymbol{x}_0 \sim p_0$ and $\boldsymbol{x}_1 \sim p_{\text{data}}$. We adopt the Optimal Transport (OT) path (linear interpolation) between $\boldsymbol{x}_0$ and $\boldsymbol{x}_1$:

$$\psi_t(\boldsymbol{x}_0, \boldsymbol{x}_1) = (1 - t)\boldsymbol{x}_0 + t\boldsymbol{x}_1, \quad t \in [0, 1]. \tag{15}$$

The corresponding (pair-conditioned) vector field is the time derivative of this path:

$$\boldsymbol{u}_t(\psi_t(\boldsymbol{x}_0, \boldsymbol{x}_1) \mid \boldsymbol{x}_0, \boldsymbol{x}_1) = \frac{d}{dt}\psi_t(\boldsymbol{x}_0, \boldsymbol{x}_1) = \boldsymbol{x}_1 - \boldsymbol{x}_0. \tag{16}$$

The Flow Matching objective regresses the model $\boldsymbol{v}_\theta$ to this vector field:

$$\mathcal{L}_{\text{FM}}(\theta) = \mathbb{E}_{t \sim \mathcal{U}[0,1], \boldsymbol{x}_0 \sim p_0, \boldsymbol{x}_1 \sim p_{\text{data}}}\left[\|\boldsymbol{v}_\theta(\psi_t(\boldsymbol{x}_0, \boldsymbol{x}_1), t) - (\boldsymbol{x}_1 - \boldsymbol{x}_0)\|^2\right]. \tag{17}$$

This formulation is equivalent to Rectified Flow (Liu et al., 2022), enabling fast ODE solver sampling.

UMMs like BAGEL (Deng et al., 2025), BLIP-3o (Chen et al., 2025a), and MetaQueries (Pan et al., 2025) adopt diffusion generation paradigms.

## A.3. Masked Autoregressive Modeling

Masked Autoregressive (Li et al., 2024b) modeling combines the sequence modeling capability of AR with the density estimation quality of Diffusion, eliminating the need for discrete codebooks.

**Continuous Factorization.** Let $\boldsymbol{z} = [\boldsymbol{z}_1, \dots, \boldsymbol{z}_T]$ be a sequence of continuous feature vectors (e.g., from a VAE encoder). The joint distribution is factorized autoregressively:

$$p_\theta(\boldsymbol{z}) = \prod_{i=1}^{T} p_\theta(\boldsymbol{z}_i \mid \boldsymbol{z}_{<i}). \tag{18}$$

Crucially, since $\boldsymbol{z}_i$ is continuous, modeling $p_\theta(\boldsymbol{z}_i \mid \boldsymbol{z}_{<i})$ with a simple unimodal loss (like MSE) results in blurry predictions. Instead, the conditional distribution is modeled using a Diffusion Loss.

**Diffusion Loss per Token.** For each step $i$, the prediction of the current token $\boldsymbol{z}_i$ is treated as a generative denoising task. We diffuse $\boldsymbol{z}_i$ to $\boldsymbol{z}_i^{(k)}$ by adding noise. The model is trained to denoise $\boldsymbol{z}_i^{(k)}$, conditioned on the clean history $\boldsymbol{z}_{<i}$:

$$\mathcal{L}_{\text{MAR}}(\theta) = \mathbb{E}_{\boldsymbol{z}, i, k, \boldsymbol{\epsilon}}\left[\|\boldsymbol{\epsilon} - \boldsymbol{\epsilon}_\theta(\boldsymbol{z}_i^{(k)}, k, \text{cond} = \boldsymbol{z}_{<i})\|^2\right]. \tag{19}$$

Here, the "head" of the autoregressive model is effectively a small diffusion model. This allows for generating high-fidelity continuous tokens sequentially without quantization artifacts.

UMMs like Harmon (Wu et al., 2025c), Fluid (Fan et al., 2025), and Skywork UniPic (Wang et al., 2025) adopt Masked Autoregressive generation paradigms.

# B. More Implementation Details

## B.1. Training Details

We present the detailed hyperparameters and training configurations for each UMM architecture in our experiments.

- *Harmon*: We use AdamW optimizer with a learning rate of 1e-5. The batch size is 16 per task (64 total per GPU across 4 tasks) with a gradient accumulation of 4 steps. The model is trained for 4,000 steps. The overall training time is approximately 5 hours.

- *OpenUni*: We use AdamW optimizer with a learning rate of 1e-5. The batch size is 4 per task (16 total per GPU across 4 tasks) with a gradient accumulation of 4 steps. The model is trained for 2,000 steps. Additionally, for OpenUni's decoupled architecture where generation and understanding are separated, we then freeze the understanding component and conduct 3,000 steps of pixel representation generation training to further improve its generation capability (taking only about 30 minutes). The overall training time is approximately 3 hours.

- *Show-o*: We use AdamW optimizer with a learning rate of 1e-6. The batch size is 2 per task (8 total per GPU across 4 tasks) with a gradient accumulation of 5 steps. The model is trained for 2,500 steps. The overall training time is approximately 8 hours.

## B.2. Depth Map Generation

We employ Depth-Anything-V2 (Yang et al., 2024) with the ViT-Large encoder to generate monocular depth estimation maps. The input images are first resized to $518 \times 518$ pixels using cubic interpolation. After inference, the predicted depth maps are resized back to the original image resolution using bilinear interpolation. The depth values are then normalized to the range $[0, 255]$ using min-max normalization:

$$D_{\mathrm{norm}} = \frac{D - D_{\mathrm{min}}}{D_{\mathrm{max}} - D_{\mathrm{min}}} \times 255 \tag{20}$$

where $D$ represents the raw depth prediction. To ensure compatibility with RGB image processing pipelines during training, the single-channel grayscale depth map is replicated across three channels.

## B.3. Segmentation Map Generation

For segmentation, we utilize the Segment Anything Model (SAM) (Kirillov et al., 2023) with the ViT-H backbone. We employ SAM's *automatic mask generator* with default parameters, sampling a uniform grid of $32 \times 32 = 1024$ point prompts across the image. Candidate masks are filtered based on predicted IoU threshold (0.88) and stability score threshold (0.95), with duplicate masks removed via non-maximum suppression (box IoU threshold 0.7). After obtaining all valid masks, we set the mask boundaries to white and the background to black. The resulting map is then converted to a 3-channel image for consistency with the training pipeline.

## B.4. Prompt Templates for UniMRG

During UniMRG training, to prevent the model from overfitting to specific prompts, we use multiple prompts with similar meanings but different phrasings. The prompts for image reconstruction are from RecA, while the prompts for image-to-depth and image-to-segmentation generation are shown below.

**Image-to-Depth Prompts:**

- "Generate the depth map of this image."

- "Produce a depth estimation for this image."

- "Compute the depth map from this photograph."

- "Extract depth information from this picture."

- "Show me the depth map."

- "What is the depth of this image?"

- "Provide the depth estimation."

- "Perform depth estimation on this image."

- "Apply monocular depth prediction."

- "Generate per-pixel depth values."

**Image-to-Segmentation Prompts:**

- "Generate the segmentation mask of this image."

- "Create a segmentation map for this image."

- "Produce a semantic segmentation mask from this image."

- "Segment this image into different regions."

- "Predict the object masks in this image."

- "Compute the segmentation from this photograph."

- "Produce an object separation mask."

- "Segment the foreground from background."

- "Segment everything in this image."

- "Generate automatic masks for all objects."

## C. Qualitative Results on Representation Generation

Figure 7 shows the depth and segmentation maps generated by different UMMs before and after UniMRG training. As can be seen, the base models of Harmon, OpenUni, and Show-o cannot generate reasonable depth or segmentation maps, with outputs resembling image reconstruction rather than structured representations. Show-o in particular fails to generate detailed information.

After UniMRG training, Harmon and OpenUni exhibit notable improvements in generating depth and segmentation maps. Specifically, both models can generate semantically correct objects from the original images. For example, in the first row, they correctly generate the two boys and the grassland from the input image. Moreover, the depth maps show that distant backgrounds (sky) are rendered in black while closer foreground objects are rendered in white, demonstrating that the models have learned to distinguish relative distances and thus enhanced their spatial understanding. Additionally, the generated representations differ from the ground truth in fine-grained details, as the input images are processed through the visual understanding encoder, which primarily captures semantic information.

For Show-o, the generated outputs are purely black when generating *depth* and *segmentation* representations, likely due to a representational bottleneck (as noted in Section 4.4): its codebook contains only 4,096 tokens, which fundamentally limits its capacity. As a result, it fails to jointly generate informative *pixel*, *depth*, and *segmentation* representations, and collapses to black outputs for *depth* and *segmentation* generation.

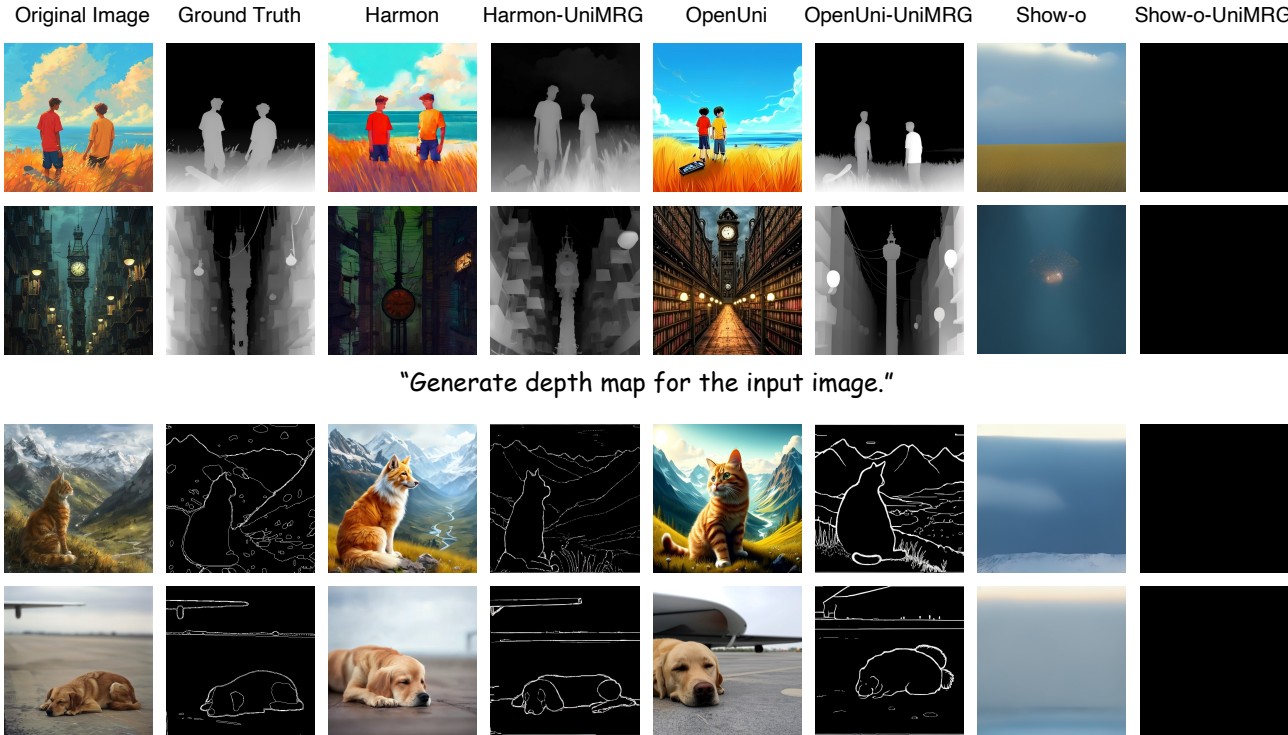

| Original Image | Ground Truth | Harmon | Harmon-UniMRG | OpenUni | OpenUni-UniMRG | Show-o | Show-o-UniMRG |

"Generate depth map for the input image."

"Generate segmentation map for the input image."

*Figure 7.* **Depth and segmentation map generation by different UMMs before and after UniMRG training.**

*Table 4.* **Detailed results on generation benchmarks.**

| Method | GenEval | | | | | | | DPGBench |
|---|---|---|---|---|---|---|---|---|
| | Single Obj. | Two Obj. | Counting | Colors | Position | Color Attri. | Overall | |
| *Harmon-1.5B* | | | | | | | | |
| Base | 99.38 | 86.11 | 66.98 | 83.87 | 45.00 | 46.92 | 71.37 | 80.52 |
| SFT | 1.77 | 0.00 | 0.00 | 0.00 | 0.00 | 0.00 | 0.30 | 2.85 |
| RecA | **100.00** | **97.98** | 69.58 | **92.82** | 74.17 | 68.58 | 83.86 | **86.61** |
| Ours | 99.90 | 97.39 | **72.71** | 89.63 | **80.42** | **71.50** | **85.26** | 85.27 |
| *OpenUni-3.6B* | | | | | | | | |
| Base | 98.65 | 73.82 | 52.50 | 79.88 | 21.92 | 40.00 | 61.13 | 79.41 |
| SFT | 96.46 | 49.41 | 21.56 | 73.85 | 13.08 | 21.83 | 46.03 | 70.24 |
| RecA | **99.17** | **87.46** | 55.00 | **84.13** | **36.08** | **58.58** | **70.07** | 81.54 |
| Ours | 98.96 | 84.01 | **55.94** | 84.04 | 29.42 | 55.67 | 68.00 | **81.78** |
| *Show-o-1.3B* | | | | | | | | |
| Base | 97.40 | 82.91 | 67.92 | 80.59 | 27.08 | 48.00 | 67.32 | 81.94 |
| SFT | 97.81 | 83.50 | 65.73 | 79.52 | 27.58 | 48.42 | 67.09 | 81.98 |
| RecA | 98.44 | **90.82** | 67.81 | 80.23 | 36.75 | **55.08** | **71.52** | 84.44 |
| Ours | **98.85** | 90.07 | **67.92** | **80.67** | **38.92** | 52.00 | 71.40 | **84.55** |

