# OpenReview forum: "Generation Enhances Understanding in Unified Multimodal Models via Multi-Representation Generation"
_ICML.cc/2026/Conference — ICML 2026 regular_

### Official Review · Reviewer_SgAS · 2026-02-14

**Soundness:** 2
**Presentation:** 3
**Significance:** 2
**Originality:** 2
**Overall Recommendation:** 4
**Confidence:** 3

**Summary:**

This paper proposes UniMRG (Unified Multi-Representation Generation), a new post-training framework that leverages auxiliary generation tasks like depth estimation, segmentation, and pixel reconstruction, in order to enhance the visual understanding capabilities of Unified Multimodal Models (UMMs). Extensive experiments across various architectures demonstrate that this method effectively bridges the gap between generation and understanding.

**Compliance With Llm Reviewing Policy:**

Affirmed.

**Final Justification:**

Thank you to the authors for their efforts and detailed responses. The responses address most of my concerns. After reviewing the clarifications, I believe the paper meets the acceptance criteria. Therefore, I’ve decided to Weak Accept.

**Key Questions For Authors:**

Please see the weaknesses.

**Limitations:**

The discussion on technical limitations is insufficient. The authors should explicitly address the reliance on uncurated pseudo-labels from off-the-shelf models (Depth Anything V2, SAM) and the potential for label noise to degrade performance in complex scenes.

**Strengths And Weaknesses:**

Strengths:

1. The motivation is well-grounded and intuitive.

2. The proposed method is resource-efficient, requiring minimal training time (e.g., approximately 3 to 8 hours on 8 H20 GPUs) to yield significant performance improvements, making it a practical for enhancing large-scale multimodal models.

3. Unlike standard SFT which often degrades generation performance by the catastrophic forgetting, UniMRG successfully achieves simultaneous improvements in both visual understanding and image generation tasks.


Weaknesses:

The core idea of leveraging generation or reconstruction losses to enhance understanding capabilities is well-established by prior work. Prior works, such as MAE [1] and QualNet [2], have demonstrated that generative objectives can improve representation learning. Therefore, the proposed method represents an incremental improvement rather than a fundamental breakthrough.

1. Despite the known risk of negative transfer between understanding and generation tasks, the paper reports improvements in both. However, it lacks a sufficient analysis of how this conflict is resolved. For example, QualNet [2] employes an invertible encoder optimized for understanding, which is then used as the generation decoder, effectively minimizing the conflict between the two tasks. Therefore, more in-depth experimentation is required to validate the synergy between these conflicting objectives.

2. Despite claims of enhanced spatial understanding and fine-grained perception, the evaluation scope is limited. The absence of standard general benchmarks (e.g., VQAv2 [3], GQA [4], MME [5]) makes it difficult to verify the model's general capabilities compared with other methods. Furthermore, the exclusion of spatial understanding benchmarks like MMMU [6], BLINK [7] and SpatialRGPT [8] restricts the comprehensive validation of the proposed geometric and structural improvements.

2. The architecture utilizes a single shared decoder for all generation tasks (pixel, depth, segmentation) without justifying this design choice against a multi-head alternative. It remains unclear whether a shared decoder is optimal. A comparative experiment with a multi-head baseline is necessary to validate the effectiveness of the proposed method.

3. The training objective assigns equal weights (lambda=1) to all loss components (pixel, depth, segmentation, understanding) without providing a rationale. The paper lacks a sensitivity analysis or ablation study on these hyperparameters to determine the optimal balance between conflicting tasks and a discussion why these parameters are good for the optimal balance between conflicting tasks.

4. The framework offers no explicit mechanism to handle or mitigate the inherent noise and inaccuracies from the "expert models" used for pseudo-labeling. The paper assumes the expert outputs are reliable enough, but it does not propose any novel technique for noise correction, uncertainty modeling, or robust data pipeline. This oversight means the model's performance is fundamentally capped by the quality of its noisy supervision. The paper lacks an analysis of how this potential noise affects the model's learning stability and final performance.

[1] He et al., "Masked Autoencoders Are Scalable Vision Learners", CVPR 2022

[2] Kim et al., "Quality-Agnostic Image Recognition via Invertible Decoder", CVPR 2021

[3] Goyal et al., "Making the v in vqa matter: Elevating the role of image understanding in visual question answering.", CVPR 2017

[4] Hudson et al., "Gqa: A new dataset for real-world visual reasoning and compositional question answering.", CVPR 2019

[5] Fu et al., "A comprehensive evaluation benchmark for multimodal large language models", arXiv:2306.13394

[6] Yue et al., "Mmmu: A massive multi-discipline multimodal understanding and reasoning benchmark for expert agi", CVPR 2024

[7] Fu et al., "Blink: Multimodal large language models can see but not perceive", ECCV 2024

[8] Cheng et al., "Spatialrgpt: Grounded spatial reasoning in vision-language models", NeurIPS 2024

---

> ### Author Rebuttal · Authors · 2026-03-28
>
> Thank you for your valuable comments and recognition of our well-grounded motivation, resource efficiency, and simultaneous improvements in both visual understanding and generation quality. We hope the following discussion can address your concerns!
>
> ---
>
> > **Q0**: The core idea of leveraging generation or reconstruction losses to enhance understanding is well-established by prior works such as MAE and QualNet.
>
> **A0**: We acknowledge that MAE, QualNet, and UniMRG all employ reconstruction-based objectives. However, MAE and QualNet leverage reconstruction-style objectives in **pixel space** to improve representation learning. In contrast, UniMRG emphasizes **intrinsic visual representations beyond appearance-level reconstruction**. UniMRG post-trains UMMs to generate these intrinsic visual representations, encouraging them to internalize geometric cues (**depth**) and structural cues (**segmentation**) that are directly beneficial for visual understanding.
>
> > **Q1**: The paper lacks sufficient analysis of how the conflict between understanding and generation tasks is resolved. Methods like QualNet address this via an invertible encoder. More in-depth experimentation is required to validate the synergy between these conflicting objectives.
>
> **A1**: We acknowledge the known risk of negative transfer between understanding and generation tasks. Unlike QualNet, which resolves this conflict via an invertible encoder, UniMRG takes an **architecture-agnostic** approach by introducing auxiliary intrinsic representation generation tasks to internalize geometric cues (depth) and structural cues (segmentation), thereby **using generation to promote understanding**, implicitly resolving the inherent conflict between the two tasks. The ablation results below validate this synergy:
>
> | Method | MMBench | MMVP | Hallusion | RWQA | VSR |
> |---|---|---|---|---|---|
> | Harmon | 50.43 | 60.00 | 46.69 | 46.67 | 60.88 |
> | + Visual Understanding & Pixel Generation | 49.50 | 61.00 | 47.11 | 49.54 | 59.00 |
> | + **Depth & Segmentation Generation** | **52.23** | **62.67** | **49.32** | **51.90** | **61.21** |
>
> *Table: Ablation on generation-understanding synergy.*
>
> These results demonstrate that introducing generation objectives for intrinsic representations (depth and segmentation) actively promotes understanding performance, rather than competing with it. For instance, spatial understanding is noticeably enhanced (RWQA: 49.54 → 51.90).
>
> We will include additional analysis about this synergy in the final version of the paper.
>
> > **Q2**: The evaluation scope is limited. Standard general benchmarks (e.g., VQAv2, GQA, MME) and spatial understanding benchmarks (e.g., MMMU, BLINK, SpatialRGPT) are absent.
>
> **A2**: Due to limited computational resources at the time of submission, we prioritized the benchmarks reported in the paper (MMBench, MMVP, Hallusion, RWQA, VSR). We agree that evaluating on a broader set of benchmarks would make our conclusions more comprehensive and solid. Our evaluation framework is based on VLMEvalKit [1], which supports GQA, MME, MMMU, and BLINK but does not support VQAv2 and SpatialRGPT. We report the results on these additional benchmarks below:
>
> | Model | MME | GQA | BLINK | MMMU |
> |---|---|---|---|---|
> | Base | 1156.90 | 58.73 | 26.25 | 34.33 |
> | SFT | 1168.54 | 58.82 | 21.46 | 36.11 |
> | RecA | 1130.72 | 58.40 | 20.67 | 35.22 |
> | UniMRG | **1182.52** | **58.90** | **30.19** | **36.56** |
>
> *Table. Evaluation on additional benchmarks on Harmon.*
>
> Under this broader evaluation, UniMRG demonstrates clear improvements in both general understanding (MME: 1156.90 → 1182.52) and spatial understanding (BLINK: 26.25 → 30.19).
>
> We will include these additional benchmarks in the final version of the paper.
>
> > **Q3**: The use of a single shared decoder for all generation tasks (pixel, depth, segmentation) lacks justification against a multi-head alternative. A comparative experiment is necessary to validate this design choice.
>
> **A3**: Most existing UMMs, including the Harmon, OpenUni, and Show-o used in our experiments, employ a single shared decoder for image generation without any multi-head design, and UniMRG does not modify the model architecture. As a result, we are unable to conduct a comparative experiment between a shared decoder and a multi-head alternative.
>
> > **Q4**: The training objective assigns equal weights to all loss components without rationale or sensitivity analysis.
>
> **A4**: Due to the 5000-character rebuttal limit,  please refer to **A2** of Reviewer `yMTA` 's response for details.
>
> > **Q5**: The framework lacks an explicit mechanism to handle noise from pseudo-label generators, and the paper provides no analysis of how this noise affects learning stability and final performance.
>
> **A5**: Due to the 5000-character rebuttal limit, please refer to **A3** of Reviewer `V9xm` 's response for details.
>
> ---
>
> [1] Vlmevalkit: An open-source toolkit for evaluating large multi-modality models. ACM MM 2024.

---

> > ### Author Rebuttal · Reviewer_SgAS · 2026-04-02
> >
> > Thank you to the authors for their efforts and detailed responses. The responses address some of my concerns. However, further clarification is needed on the following point. The paper and response claim that generation "promotes" understanding. However, this is a observation, not a structural discussion. While the known risk of negative transfer between understanding and generation tasks, the paper needs a more rigorous and comprehensive analysis for how it minimizes the conflict between the generation and understanding tasks and why the proposed generation task helps improve understanding.

---

> > > ### Author Response · Authors · 2026-04-02
> > >
> > > We provide a more comprehensive discussion and additional experiments as follows.
> > >
> > > > **1. Where does the risk of negative transfer between understanding and generation tasks come from?**
> > >
> > > The risk mainly comes from the different requirements that the two tasks impose on the shared representation. Image generation typically requires preserving appearance-level details such as texture and color, whereas visual understanding mainly relies on semantic information. This difference in supervision focus can introduce negative transfer when the two tasks are jointly optimized on shared parameters.
> > >
> > > > **2. How does UniMRG minimize the conflict between the generation and understanding tasks?**
> > >
> > > Our claim is **not** that any generation objective will naturally promote understanding. Rather, only generation targets that are **aligned with understanding-relevant factors are beneficial**. Depth encodes geometric cues and relative distance, while segmentation emphasizes object boundaries and structural layout. These factors are directly useful for visual understanding.
> > >
> > > From this perspective, UniMRG **minimizes the conflict between the generation and understanding tasks at the supervision level** by introducing auxiliary intrinsic targets that are more closely aligned with the factors required by understanding. This is different from QualNet. QualNet addresses the conflict mainly through architectural disentanglement via an invertible decoder, while UniMRG minimizes the conflict from the supervision side by **choosing intrinsic generation tasks whose optimization directions are more compatible with understanding**.
> > >
> > > To further verify this point, we analyze the cosine similarity between gradients of different losses on the shared trainable parameters used by both generation and understanding. The results are shown below:
> > >
> > > |  | $ \cos(\nabla L_{und}, \nabla L_{pixel}) $ | $ \cos(\nabla L_{und}, \nabla L_{depth}) $ | $ \cos(\nabla L_{und}, \nabla L_{seg}) $ |
> > > |---|---:|---:|---:|
> > > | Value | -0.0474 | 0.0458 | 0.0760 |
> > >
> > > *Table. Gradient cosine similarity on the shared trainable parameters.*
> > >
> > > The absolute values are small, which is expected when cosine similarity is computed in high-dimensional parameter spaces. We find that pixel reconstruction is conflicting with understanding (-0.0474), whereas depth (0.0458) and segmentation (0.0760) show positive alignment. The sign change from negative to positive supports our claim that **intrinsic generation targets are more compatible with understanding than generic RGB reconstruction**.
> > >
> > > This is also consistent with the ablation results in Table 3 of our paper: pixel generation mainly enhances generation ability but does not improve understanding, whereas intrinsic representation generation brings clear gains on understanding benchmarks.
> > >
> > > > **3. Why does the proposed generation task help improve understanding?**
> > >
> > > In UniMRG, when performing generation tasks, the input image is first processed by the understanding encoder. In other words, visual understanding and visual generation share the understanding encoder and the LLM backbone. Therefore, the auxiliary generation losses can directly shape the shared internal representation used for downstream QA.
> > >
> > > More importantly, **QA supervision is sparse**: each image-question pair provides only a single answer target, which cannot explicitly enforce the model to capture scene geometry or object boundaries, even though these factors are highly useful for understanding. In contrast, **depth and segmentation provide dense supervision**, which fills this gap by encouraging the model to encode geometric cues (depth) and structural cues (segmentation) into its internal representation.
> > >
> > > To further validate this mechanism, beyond the improvements on multiple understanding benchmarks, we conduct an additional probing experiment to test whether the intermediate features of the UMM contain more depth information after UniMRG training. Specifically, we feed an image into a frozen UMM (Harmon), extract the intermediate features of UMM, randomly sample two spatial positions, and use a single learnable linear layer to predict which position is closer. The probing results are shown below:
> > >
> > > | Model | Depth probing accuracy |
> > > |---|---:|
> > > | Base | 69.5 |
> > > | UniMRG | **81.2** |
> > >
> > > *Table. Depth probing results on the intermediate features of UMM.*
> > >
> > > The result suggests that, after UniMRG training, the intermediate features of the UMM contain more depth information, leading to higher probing accuracy. This provides additional evidence that **UniMRG improves understanding by reshaping the internal representation toward understanding-relevant geometric and structural information**.
> > >
> > > We thank the reviewer again for raising this important point. We will incorporate this clearer discussion and the additional evidence into the final version of the paper. If there are any remaining concerns, we would be happy to further clarify them.

---

### Official Review · Reviewer_yMTA · 2026-03-09

**Soundness:** 3
**Presentation:** 3
**Significance:** 2
**Originality:** 2
**Overall Recommendation:** 4
**Confidence:** 4

**Summary:**

This paper proposes to enhance visual understanding performance in unified multimodal models by enforcing the model to generate depth and segmentation masks along with images. The approach they use is post-training that takes images (pixel-depth-segmentation) and question-answer pairs as input to finetune UMMs. The pixel-depth-segmentation reconstruction objectives encourages UMMs to encode not only texture or color but geometric and structure information of images into representations. Containing rich high-level information, these representations enables more accurate scene understanding in terms of spatial relations or structural layouts.

**Compliance With Llm Reviewing Policy:**

Affirmed.

**Key Questions For Authors:**

See above.

**Limitations:**

Yes

**Strengths And Weaknesses:**

**Strength**

1. The paper is well-written and easy to follow.

2. The idea is straightforward that further incorporates depth and segmentation reconstruction loss on top of training objectives of UMMs.

3. Extensive experiments are conducted on different UMM architectures, and the proposed approach outperforms baselines.

**Weakness**

1. Improvements are relatively small (especially when comparing with pixel reconstruction + visual understanding case), which are at the cost of additional preprocessing like generating depth maps or generating segmentation maps.

2. The proposed training objective is similar to multi-task learning, yet the paper presents it mainly as a generation-enhances-understanding mechanism without discussing its connection to MTL. Task weighting strategies is not explored; all objective weights are set to 1 without justification.

3. For OpenUni-3.6B, the RecA performs EXACTLY same as the base model for understanding tasks?

---

> ### Author Rebuttal · Authors · 2026-03-28
>
> Thank you for your valuable comments and recognition of the clarity of our paper, the straightforward design, and the extensive cross-architecture experiments demonstrating consistent improvements. We hope the following discussion can address your concerns!
>
> ---
>
> > **Q1**: Improvements are relatively small (especially when comparing with pixel reconstruction + visual understanding case), which are at the cost of additional preprocessing like generating depth maps or generating segmentation maps.
>
> **A1**: While the improvements may appear moderate in certain settings, they are **consistently observed across multiple UMM architectural paradigms**, demonstrating the generality and robustness of our approach.
>
> Regarding the preprocessing cost, we emphasize that it is a **one-time cost** that does not need to be repeated across training runs. More importantly, **preprocessing is not a core component of UniMRG and is replaceable**. Comparable results can be achieved by directly using existing real-world depth datasets without any pseudo-label generation. To demonstrate this, we replaced the pseudo-labeled depth maps with the real-world depth dataset NYU V2 [1] and report the results below:
>
> | Model | MMBench | MMVP | Hallusion | RWQA | VSR | GenEval | DPG |
> |---|---|---|---|---|---|---|---|
> | Base | 50.43 | 60.00 | 46.69 | 46.67 | 60.88 | 71.37 | 80.52 |
> | UniMRG (pseudo-labeled depth) | 52.23 | 62.67 | 49.32 | 51.90 | 61.21 | 85.26 | 85.27 |
> | UniMRG (real-world depth) | 52.15 | 63.33 | 48.48 | 51.76 | 61.37 | 85.75 | 85.34 |
>
> *Table. Comparison of pseudo-labeled depth vs. real-world depth (NYU V2) on Harmon.*
>
> The results show that replacing pseudo-labeled depth with NYU V2 leads to only marginal differences, indicating that **preprocessing is a replaceable component in UniMRG and can be substituted with off-the-shelf real-world datasets at no additional preprocessing cost**.
>
> We thank the reviewer for raising this point, and we will include a discussion of preprocessing costs in the final version of the paper.
>
> > **Q2**: The proposed training objective is similar to multi-task learning, yet the paper presents it mainly as a generation-enhances-understanding mechanism without discussing its connection to MTL. Task weighting strategies is not explored; all objective weights are set to 1 without justification.
>
> **A2**:  Yes, we acknowledge the similarity to MTL. While both share a multi-objective optimization form, UniMRG differs in purpose: rather than optimizing multiple tasks for their individual performance as in MTL, UniMRG learns different visual representations of an image to help the model understand visual content from multiple perspectives.
>
> Regarding the task weighting strategy, we set all objective weights to 1 in the interest of simplicity. We acknowledge that exploring task weighting strategies is a valid concern, and we have conducted additional experiments to address this.
>
> In practice, our method is **not highly sensitive to the choice of weights**. We varied the weights by setting each of the four terms to 0.5 in turn while keeping the others at 1, and compared against the all-ones setting. The results are as follows:
>
> | Model | MMBench | MMVP | Hallusion | RWQA | VSR | GenEval | DPG |
> |---|---|---|---|---|---|---|---|
> | Base | 50.43 | 60.00 | 46.69 | 46.67 | 60.88 | 71.37 | 80.52 |
> | UniMRG (all weights = 1) | 52.23 | 62.67 | 49.32 | 51.90 | 61.21 | 85.26 | 85.27 |
> | $\lambda_{\text{pixel}}=0.5$ | 51.37 | 62.67 | 48.26 | 51.37 | 61.87 | 85.20 | 83.09 |
> | $\lambda_{\text{seg}}=0.5$ | 51.98 | 63.00 | 48.26 | 51.63 | 61.37 | 85.49 | 83.99 |
> | $\lambda_{\text{depth}}=0.5$ | 52.06 | 63.67 | 48.90 | 52.03 | 61.29 | 85.33 | 83.92 |
> | $\lambda_{\text{und}}=0.5$ | 52.15 | 63.00 | 48.37 | 51.11 | 61.37 | 83.67 | 84.96 |
>
> *Table. Sensitivity analysis on task weights on Harmon.*
>
> The results show that different weight configurations yield similar performance, confirming that **UniMRG is robust to the choice of task weights**.
>
> Nevertheless, **each task objective plays an indispensable role, as validated by the ablation study in Table 3 of our paper.** Starting from the base Harmon model: adding the visual understanding loss improves understanding but degrades generation quality; adding pixel reconstruction then significantly enhances generation quality; finally, incorporating depth and segmentation representation generation further strengthens fine-grained perception, mitigates hallucinations, and improves spatial understanding.
>
> We will include a discussion of MTL and task weighting in the final version of the paper.
>
> > **Q3**: For OpenUni-3.6B, the RecA performs EXACTLY same as the base model for understanding tasks?
>
> **A3**: Yes, this is expected. In the RecA training paradigm, the understanding module of OpenUni is kept frozen, so the understanding performance remains identical to that of the base model by design.
>
>
> ---
>
> [1] Indoor Segmentation and Support Inference from RGBD Images. ECCV 2012.

---

> > ### Author Rebuttal · Reviewer_yMTA · 2026-04-03
> >
> > Thanks for the rebuttal. I keep my rating as the original.

---

> > > ### Author Response · Authors · 2026-04-03
> > >
> > > Thank you for your feedback and for the constructive review process. We sincerely appreciate your time, consideration, and valuable insights.

---

### Official Review · Reviewer_V9xm · 2026-03-14

**Soundness:** 3
**Presentation:** 3
**Significance:** 3
**Originality:** 3
**Overall Recommendation:** 4
**Confidence:** 3

**Summary:**

This paper studies the reverse direction of the now-common "understanding improves generation" paradigm for unified multimodal models. Instead of using understanding signals to improve synthesis, the paper proposes UniMRG, an architecture-agnostic post-training method that improves visual understanding by training the model to generate multiple intrinsic visual representations of an input image, specifically pixel reconstruction, depth, and segmentation, alongside standard understanding objectives. The intuition is that these auxiliary generation targets encourage the model to internalize appearance, geometry, and structure in a way that transfers to downstream understanding. The evaluation covers several UMM architectures and reports gains on fine-grained perception, hallucination reduction, spatial understanding, and some generation metrics.

**Compliance With Llm Reviewing Policy:**

Affirmed.

**Key Questions For Authors:**

1. How much of the downstream gain comes from simply adding more generation supervision, versus specifically adding depth and segmentation as intrinsic targets?
2. What is the preprocessing and post-training cost of obtaining the depth/segmentation targets and training UniMRG, relative to a simpler understanding-only SFT baseline?
3. Have the authors tested how sensitive the gains are to the quality of the pseudo-target generators, especially for segmentation and depth under domain shift?

**Limitations:**

The paper discusses the capacity bottleneck observed in Show-o, but it should more explicitly discuss dependence on pseudo-target quality and the practical training cost of the recipe.

**Strengths And Weaknesses:**

**Strengths**
- Using auxiliary generation of pixel, depth, and segmentation targets to improve understanding is a clean and practically useful recipe.
- The cross-architecture evaluation helps the paper. Testing several unified multimodal models makes the claim of generality more credible than a single-model study would.
- The empirical section is solid overall, covering fine-grained perception, hallucination reduction, spatial understanding, generation quality, and an OOD depth-generation test on MidjourneyV6.
- The ablation study is informative as well: depth and segmentation help, but pixel supervision still appears important for preserving generation quality.

**Weaknesses**
- Conceptually, this feels more like a strong post-training recipe than a major step in model design; once the reverse direction is posed, the method is fairly natural.
- The paper relies on pseudo-targets for depth and segmentation, yet the dependence on target quality and possible bias from those generators is not analyzed carefully.
- The gains are not equally convincing across architectures, with Show-o benefiting much less than the other model families.

---

> ### Author Rebuttal · Authors · 2026-03-28
>
> Thank you for your valuable comments and recognition of the clean and practical recipe, the cross-architecture evaluation, the solid empirical coverage, and the informative ablation study. We hope the following discussion can address your concerns!
>
> ---
>
> > **Q1**: How much of the downstream gain comes from simply adding more generation supervision, versus specifically adding depth and segmentation as intrinsic targets?
>
> **A1**: The ablation study in Table 3 of the paper isolates the contribution of adding depth and segmentation as intrinsic targets. As shown below:
>
> | Method | MMBench | MMVP | Hallusion | RWQA | VSR |
> |---|---|---|---|---|---|
> | Harmon | 50.43 | 60.00 | 46.69 | 46.67 | 60.88 |
> | + Visual Understanding & Pixel Generation | 49.50 | 61.00 | 47.11 | 49.54 | 59.00 |
> | + **Depth & Segmentation Generation** | **52.23** | **62.67** | **49.32** | **51.90** | **61.21** |
>
> *Table: Ablation study on intrinsic generation targets.*
>
> Adding depth and segmentation as intrinsic targets yields clear and consistent improvements in understanding capabilities. For instance, hallucination mitigation improves notably (Hallusion: 47.11 → 49.32), and spatial understanding is significantly enhanced (RWQA: 49.54 → 51.90).
>
>
> > **Q2**: What is the preprocessing and post-training cost of obtaining the depth/segmentation targets and training UniMRG, relative to a simpler understanding-only SFT baseline?
>
> **A2**: We would first like to highlight that an understanding-only SFT baseline causes **a severe degradation in the generation capability of UMMs**. As shown in Table 1 of our paper, applying SFT to Harmon causes GenEval to drop drastically from 71.37 to 0.30. In contrast, UniMRG not only achieves greater improvements in understanding than the SFT baseline, but also **significantly enhances generation quality**, with GenEval improving from 71.37 to 85.26.
>
> We now discuss the preprocessing and post-training costs in detail below, all measured on 8 NVIDIA H20 GPUs.
>
> **Preprocessing Cost.** Generating depth maps takes approximately 23 hours and segmentation maps approximately 27 hours. Importantly, this is a **one-time cost** that requires no repeated iteration across training runs. We also note that **image preprocessing is not a core component of UniMRG and is replaceable. Similar results can be achieved using existing real-world datasets without any preprocessing**, which we discuss further in Q3.
>
> **Post-training Cost.** Our method is **resource-efficient**: training takes approximately 3 hours for OpenUni, 5 hours for Harmon, and 8 hours for Show-o.
>
> We thank the reviewer for raising this point, and we will include a discussion of both preprocessing and post-training costs in the final version of the paper.
>
>
> > **Q3**: Have the authors tested how sensitive the gains are to the quality of the pseudo-target generators, especially for segmentation and depth under domain shift?
>
> **A3**: Thank you for raising this point. We acknowledge that a discussion of pseudo-target quality is missing from the paper.
>
> In practice, UniMRG is not highly sensitive to the quality of the pseudo-target generators and the quality of existing SOTA generators is already sufficient. To validate this, we replaced the pseudo-labeled depth maps with a real-world depth dataset, NYU V2 [1], and report the results below:
>
>
> | Model | MMBench | MMVP | Hallusion | RWQA | VSR | GenEval | DPG |
> |---|---|---|---|---|---|---|---|
> | Base | 50.43 | 60.00 | 46.69 | 46.67 | 60.88 | 71.37 | 80.52 |
> | UniMRG (pseudo-labeled depth) | 52.23 | 62.67 | 49.32 | 51.90 | 61.21 | 85.26 | 85.27 |
> | UniMRG (real-world depth) | 52.15 | 63.33 | 48.48 | 51.76 | 61.37 | 85.75 | 85.34 |
>
> *Table. Comparison of pseudo-labeled depth vs. real-world depth (NYU V2) on Harmon.*
>
>
> The results show that replacing pseudo-labeled depth with NYU V2 leads to only marginal differences, indicating that the quality of our pseudo-labels is already sufficient.
>
> Furthermore, as shown in Figure 1 of our paper, the depth maps generated by UMMs after UniMRG post-training are not highly precise, with some differences remaining between the generated depth maps and the ground truth. This suggests that generating highly precise depth maps may be challenging for UMMs, and therefore UniMRG does not impose strong requirements on pseudo-label quality. Nevertheless, UMMs still benefit substantially from the intrinsic representation generation task, resulting in improved understanding capabilities.
>
> Regarding domain shift, we discuss this in Figure 6 of our paper. Although UniMRG is trained on real images, a clear improvement in depth estimation is still observed on generated images, demonstrating that the model has genuinely learned the spatial relationships encoded in depth maps rather than overfitting to the training distribution.
>
> We will include a discussion of pseudo-label quality in the final version of the paper.
>
> ---
>
> [1] Indoor Segmentation and Support Inference from RGBD Images. ECCV 2012.

---

> > ### Author Rebuttal · Reviewer_V9xm · 2026-04-05
> >
> > The authors addressed my concern, so I remain positive about the paper and keep my score at 4.

---

> > > ### Author Response · Authors · 2026-04-05
> > >
> > > We are glad that our response has adequately addressed your concerns about the submission. Thanks a lot for your comments and suggestions to improve our paper.

---

### Official Review · Reviewer_jTYP · 2026-03-16

**Soundness:** 2
**Presentation:** 2
**Significance:** 2
**Originality:** 2
**Overall Recommendation:** 3
**Confidence:** 4

**Summary:**

This work proposes a new post-training method on unified multimodal models (UMMs) to enhance their performances. The method involves finetuning the base model with multiple intrinsic images including pixels, depth, and segmentation maps. The authors claim that this approach can significantly improve the performance of UMMs on the understanding of visual content.

**Compliance With Llm Reviewing Policy:**

Affirmed.

**Final Justification:**

After the rebuttal, my concerns have been solved by the authors. However, the proposed method is too simple and is similar to previous literatures. Therefore, I'd rather treat this work an engineering work.

**Key Questions For Authors:**

Please see weaknesses.

**Limitations:**

Please include limitations and social impact statements in the paper.

**Strengths And Weaknesses:**

# Strengths
1. The proposed method is simple and can be easily implemented on existing UMMs without requiring significant changes to the model architecture.
2. The use of multiple intrinsic images can provide richer information for the model, which can lead to better performance in understanding visual content.
3. The authors provide experimental results that demonstrate the effectiveness of their method, showing significant improvements in performance on various benchmarks.

# Weaknesses
1. The proposed method is not novel. Specifically, it's recommended that the authors discuss the pap er, "Chain-of-Visual-Thought: Teaching VLMs to See and Think Better with Continuous Visual Tokens", which also uses multiple intrinsic images to enhance the performance of visual language models. Please note that although the CoVT paper focuses on visual language models, while UniMRG targets unified multimodal models, the underlying idea of using multiple intrinsic images is similar. Therefore, it is important for the authors to acknowledge and discuss the CoVT paper.
2. In figure 4, the authors show several examples of the model's performance after finetuning. Notably, many of the examples are general understanding cases. It's counterintuitive that the model's ability to understand general visual content is significantly improved after finetuning with intrinsic images. The authors should explain more clearly why finetuning with intrinsic images can lead to such significant improvements in general visual understanding. Does the improvement come from overfitting to the specific examples used for finetuning? I don't see a satisfactory explanation in section 4.4.

---

> ### Author Rebuttal · Authors · 2026-03-28
>
> Thank you for your valuable comments and recognition on the simplicity and easy implementation of our method on existing UMMs, the effectiveness of leveraging multiple intrinsic images for richer visual understanding, and the significant performance improvements demonstrated across various benchmarks. We hope the following discussion can address your concerns!
>
> ---
>
> > **Q1**: It's recommended that the authors discuss the paper, "Chain-of-Visual-Thought: Teaching VLMs to See and Think Better with Continuous Visual Tokens", which also uses multiple intrinsic images to enhance the performance of visual language models.
>
> **A1**: We thank you for pointing out this relevant work that we overlooked. Per the ICML 2026 policy, *"Authors are not required (but still welcome) to discuss works that have been made public less than two months before the full-paper submission deadline. Such recent papers should be considered as concurrent and simultaneous. Good judgment is necessary to decide whether a paper that has not yet been peer-reviewed should be discussed."*  We therefore treat CoVT [1] as a concurrent work. Nevertheless, we are happy to discuss the differences between CoVT and UniMRG. The two works differ in motivation, training paradigm, and inference behavior.
>
> (1) **Different motivation.** CoVT is motivated by improving **VLM reasoning** by explicitly introducing continuous visual tokens into the CoT process, so that intrinsic cues can serve as part of the **reasoning chain**. In contrast, UniMRG is motivated by a different question: **can generation improve understanding in UMMs?** Our goal is not to redesign the reasoning interface, but to use multi-representation generation as auxiliary supervision so that geometric and structural regularities are internalized into the **model’s internal representation space** and improve downstream understanding.
>
> (2) **Different training paradigm.** CoVT adopts a four-stage training pipeline and introduces additional components (e.g., PIDINet Encoder) for continuous visual token learning, with manually specified token counts. UniMRG instead uses a simple, architecture-agnostic post-training recipe. **UniMRG does not require multi-stage training or modifications to the UMM architecture**, making it easier to integrate across different UMM families.
>
> (3) **Different inference behavior.** In CoVT, continuous visual tokens are involved in the reasoning process at inference time. In UniMRG, inference remains exactly the same as in the original UMM, **without additional inference overhead**.
>
> We will add a discussion of CoVT in the final version to clarify these differences. We thank the reviewer again for bringing this to our attention.
>
>
> > **Q2**: In Figure 4, the authors show several examples of the model's performance after finetuning. Notably, many of the examples are general understanding cases.
>
> **A2**: Thank you for pointing this out. We apologize for the unclear description in Figure 4, which may have caused this misunderstanding.
>
> In fact, the examples shown are not general visual understanding examples. Specifically, the first row **Fine-grained Perception** contains examples from the fine-grained understanding benchmark *MMVP* [2]; the second row **Hallucination** contains examples from the hallucination benchmark *HallusionBench* [3]; and the third row **Spatial Understanding** contains examples from spatial understanding benchmarks, with the first two drawn from *RealWorldQA* [4] and the third from *Visual Spatial Reasoning* [5].
>
> These results demonstrate that by learning the geometric and structural regularities through the intrinsic representation generation task, UniMRG effectively enhances several specific understanding capabilities of UMMs, including fine-grained perception, hallucination mitigation, and spatial understanding.
>
> We will clarify the caption of Figure 4 more explicitly in the final version of the paper to avoid such confusion.
>
> > **Q3**: Please include limitations and social impact statements in the paper.
>
> **A3**: Thank you for pointing out this unclear aspect of our paper. The limitations are discussed in Section 4.4 (Discussion), and the social impact is presented in the Impact Statement on page 9.
>
> In the final version, we will move the limitations into a dedicated section with its own heading for improved clarity and visibility.
>
> ---
>
> [1] Chain-of-Visual-Thought: Teaching VLMs to See and Think Better with Continuous Visual Tokens. Arxiv 2025.
>
> [2] Eyes wide shut? exploring the visual shortcomings of multimodal llms. CVPR 2024.
>
> [3] Hallusionbench: an advanced diagnostic suite for entangled language hallucination and visual illusion in large vision-language models. CVPR 2024.
>
> [4] Grok-1.5 Vision Preview, 2024.
>
> [5] Visual spatial reasoning. TACL 2023.

---

> > ### Author Rebuttal · Reviewer_jTYP · 2026-04-03
> >
> > The rebuttal has resolved my concens. Therefore, I'll raise my score.

---

> > > ### Author Response · Authors · 2026-04-03
> > >
> > > Thank you very much for revisiting our rebuttal and for raising your score. We sincerely appreciate your time and consideration.
> > >
> > > We are glad to know that our rebuttal has adequately addressed your concerns. Since the current rating is still a weak reject, we were wondering whether it fully reflects your updated assessment. If there are any remaining issues that we have not yet fully addressed, we would be very grateful for the opportunity to clarify them further.
> > >
> > > Thank you again for your helpful feedback and careful review.

---

### Decision · Program_Chairs · 2026-04-30

**Decision:**

Accept (regular)

**Comment:**

This work was accepted for publication because it proposes a reverse approach “using generative tasks to improve understanding capabilities” in a multimodal model (UMM) that integrates visual understanding and generation, and demonstrated the effectiveness of this approach.

Although the paper was accepted, reviewers pointed out that further exploration and revisions are needed in the following areas. Please incorporate these comments into the camera-ready version.

Since the method of “improving representation learning using generation loss” is already well-established (e.g., MAE and CoVT), please more clearly define the specific differences from these prior studies and the unique contributions of UniMRG. Although the outputs (pseudo-labels) of the “expert model” are used for depth and segmentation generation, an analysis of the impact of noise in these labels on learning, as well as an examination of mechanisms to mitigate it, is desirable. While the weights of each loss function (pixel, depth, etc.) are all set to 1, a sensitivity analysis to avoid task competition and a detailed discussion regarding the optimal balance are necessary. Additionally, further evaluations should be conducted on standard general-purpose benchmarks that assess spatial understanding (e.g., SpatialRGPT) and domain-specific knowledge (e.g., MMMU) to demonstrate the model’s versatility from multiple perspectives. Furthermore, since the extent of improvement varies across models (e.g., Show-o), a detailed analysis is needed on how a model’s capacity and characteristics influence the effectiveness of UniMRG.